# How to Improve the Total Cost of Ownership of Electric Vehicles: An Analysis of the Light Commercial Vehicle Segment

**Philippe Lebeau [1],*, Cathy Macharis [1] and Joeri Van Mierlo [1,2]**

[1] Mobility, Logistics and Automotive Technology Research Centre (MOBI) Research Group, Vrije Universiteit Brussel, Pleinlaan 2, 1050 Brussels, Belgium; Cathy.Macharis@vub.be (C.M.); Joeri.Van.Mierlo@vub.be (J.V.M.)

[2] Flanders Make, 3001 Heverlee, Belgium

\* Correspondence: philippe.Lebeau@vub.be; Tel.: +32-2-614-83-24

**Abstract:** This paper analyses how the total cost of ownership (TCO) of electric light commercial vehicles change with the number of kilometers driven, the period of ownership, the residual value of the battery, and different fiscal incentives, as well as a kilometer charging scheme. This paper demonstrates that a kilometer-based charge and reduced fiscal incentives for conventional vans can drastically improve the TCO of electric commercial light duty vehicles. Second life applications for batteries could also have a strong impact on the TCO of electric vans as they could retrieve a better residual value. Finally, the paper shows that the TCO of electric vans can be optimized based on its usage. These are important findings given the ambitious objective of carbon free city logistics by 2030. Adoption of electric vans remains very low and this paper offers an up to date analysis to stimulate the electrification of light commercial vehicles, a segment that is growing fast in city logistics.

**Keywords:** total cost of ownership; light commercial vehicles; city logistics

## 1. Introduction

Climate change is coming at the top of the political agenda. The recent report of the IPCC GIEC reminded the importance of acting now if we want to limit impacts of global warming to 1.5 degrees Celsius above pre industrial levels [1]. Transportation has a key role to play as it is responsible for about one fourth of GHG emissions [2]. Electrification of transport in that context can be a part of the solution. As a result, EU member states have used different policies to stimulate the adoption of electric vehicles and we can observe a progressive adoption across Europe [3].

The electrification of vans and trucks is however progressing at a slower pace. In Belgium, we do not see a similar evolution in the freight segment than in the passenger segment as shown in Figure 1. Yet, it is one of the most polluting segments of the transport sector. While freight is responsible for about 10–15% of the vehicle kilometers in cities, vans and trucks generate up to 25% of $CO_2$ emissions and 50% of $NO_x$ emissions [4]. This result can be explained by the large share of diesel in the light commercial vehicle segment. In Belgium, 92.4% of vans run with diesel [5]. Given the opportunities in terms of environmental performance of electric vehicles [6], shifting from diesel to electric vans can support the ambitious objectives set by the European Commission. We need indeed to reduce by 60% the GHG emissions generated by the transport sector by 2050 compared to the levels of 1990 [7]. And major urban centers should run with $CO_2$ free city logistics by 2030 [7].

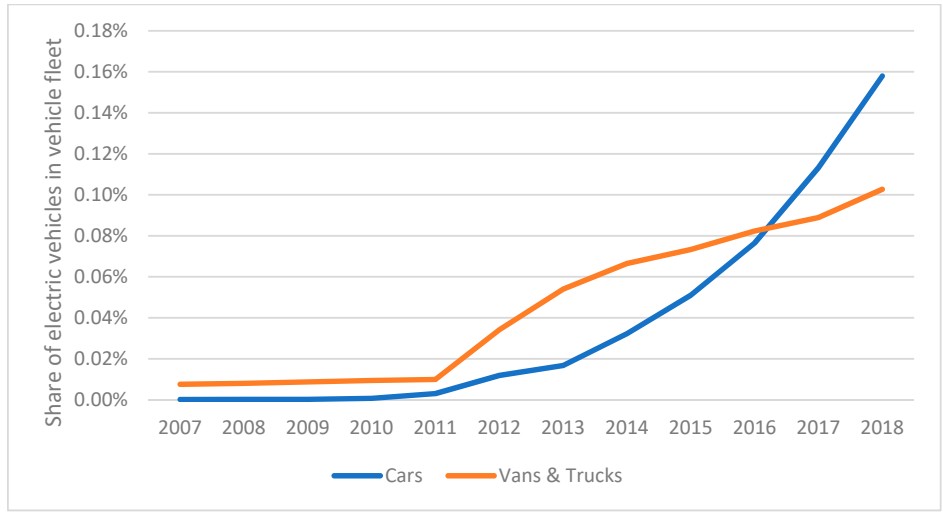

**Figure 1.** Evolution of electric vehicles' share in the vehicle fleet in Belgium [5].

Costs are often considered by companies to be a major barrier to electric vehicle adoption [8]. However, the important purchase price of electric vehicles can be partially compensated by their low running costs and maintenance costs compared to conventional vehicles. The costs structure of electric vehicles is different from conventional vehicles, therefore the total cost of ownership has often been used to compare these different technologies [9–11]. A few total cost of ownership (TCO) analyses have investigated the competitive position of commercial vehicles. These are however mainly focused on the truck segment [12–16]. Conversely, the light commercial vehicle segment remains overlooked while it was identified as a key segment for several key reasons [17]: the fleet of vans is much larger than the truck fleet and keeps growing faster, vans are especially used in last miles and urban areas, and environmental impact per ton transported is especially high with vans.

The most recent TCO analyses are unfortunately limited to the analysis of one specific electric van compared with its conventional version which does not reflect the diversity of that segment [18,19]. We need to go back to 2015 to have complete total costs of ownership analysis for that van segment [20]. Since then, the technology and the market have progressed fast. Prices of batteries have dropped. Diesel prices have increased. Regulations are changing. This paper presents therefore an up-to-date overview of the competitive position of electric vehicles in the light commercial vehicle segment of the Brussels-Capital Region. To extrapolate the results beyond Brussels, a sensitivity analysis will show how the results can change with a different policy context.

## 2. Materials and Methods

### 2.1. The Total Cost of Ownership (TCO)

Owning and operating a vehicle is associated with costs that occur at different moments in time. To be able to compare these costs across time, the total cost of ownership methodology uses the financial formula of the present discounted value. This way, every cost can be included in one cost indicator to describe the full cost of one alternative. The total cost of ownership is defined as "a purchasing tool and philosophy which is aimed at understanding the true cost of buying a particular good or service from a particular supplier" (p. 1, [21]). It gives the total discounted cost of owning, operating, and maintaining an asset over a limited period of time. It is used to compare competing investments and evaluate the most profitable alternative.

To calculate the present value of future one-time costs, the following formula is used [22]:

$$PV = A_t \times \frac{1}{(1+I)^t} \tag{1}$$

where:

*PV* = present value
$A_t$ = amount of one-time cost at a time *t*
*I* = real discount rate
*t* = time (expressed in number of years)

In general, the total cost of ownership is calculated in three steps:

- Analysis of every stream of periodic costs;
- Calculation of the present value of the one-time and the recurring costs;
- Division of the present value by the number of kilometers during the vehicle lifetime in order to compute a cost per kilometer.

## 2.2. Assumptions of the Model

Given its definition, the TCO equation can be divided into three variables: (1) the costs of ownership, (2) the period of time over which these costs occurred, and (3) the discount rate applied to future costs to actualize them.

### 2.2.1. Period of Ownership

Light commercial vehicles have an average ownership period of 10.9 years in Europe [23]. That period changes across EU member states going from 7.4 years in Germany to 17.1 years in Greece. In Belgium, the average period of ownership of light commercial vehicles was 8.2 years in 2016 [23]. We assume therefore in our model that light commercial vehicles are used for 8 years before they are sold. We will test the impact of that assumption in the sensitivity analysis.

### 2.2.2. Discount Rate

The discount rate can be defined as "the rate of interest reflecting the investor's time value of money" (p. 6, [22]). It can be either a real discount rate (excluding inflation) or a nominal discount rate (including inflation). The real discount rate has the advantage of eliminating complex accounting for inflation within the present value equation. Hence, this study uses first the long-term interest rate of state bonds to estimate the nominal discount rate. It minimizes the risk factor of the financial markets and estimates best the time value of money. For this TCO calculation, we use an interest rate of 0.51%, which is the average rate between August 2018 and August 2019 of Belgian bounds at 10 years [24]. We extract from this nominal discount rate the 1.8% of expected inflation in Belgium on the period 2021–2024 [25] to find a real discount rate of −1.29%. This negative rate reflects the limited growth and the higher inflation we expect in Belgium. Investing is therefore stimulated in such an economic climate.

### 2.2.3. Cost of Ownership

The analysis of the cost of ownership considers every cost associated to the use of the vehicle. Only investments in charging infrastructure are not included since they will be diluted according to the size of the fleet. The following costs flows are considered: road taxes, governmental support and fiscal incentives, battery, maintenance, car inspection, insurance, fuel (and electricity), and purchase costs. All costs are excluding VAT. The following assumptions of the model are applied to these costs:

In 2015, there were almost 680,000 light commercial vehicles in Belgium driving 10.97 milliard vehicle kilometers [26]. We assume therefore that light commercial vehicles drive an average of 16,000 km per year. Still we will test the impact of that assumption in the sensitivity analysis.

The insurance costs were calculated for a company with a frequent use of the vehicle, based in Brussels (postcode 1000) with no accidents in the last 5 years. The insurance is limited to the civil liability (Data collected from the insurance company Axa). No cost difference as such is applied

between electric and conventional vehicles but differences in the power of the motors may generate a variation in the insurance premiums between the different drive trains.

Maintenance costs include costs for small and large maintenance. They are different between conventional and electric vehicles. Maintenance costs of electric vehicles are more limited than conventional since they do not have an internal combustion engine: they have less moving components; they face less temperature stress, and do not need oil and filter replacements [27]. Palmer et al. (2018) reported different maintenance costs for electric, petrol, and diesel vehicles in different countries of the world [9]. They usually remain stable across countries. Costs of the closest country was therefore considered, the UK. Based on their estimation, we assume maintenance costs per year of 306 €, 397 €, and 832 € for respectively electric, petrol, and diesel vehicles (we consider here the exchange rate of the 03 October, 2019 from GBP to EUR at 1.12175).

As the vehicle is assumed to be sold on the second-hand market, its residual value is retrieved. The analysis considers an annual depreciation rate of 20.75% on the value of a diesel, 17.27% on the value of a petrol, and 15.24% on the value of an electric vehicle. These figures were computed based on the average residual values after five years published by ING Economics Department [28], assuming a constant depreciation rate.

In order to have a clear idea of the cost structure, the costs of the new battery included in the initial purchase costs are deduced from the purchase costs and affected to the battery costs category. It represents indeed a significant share of the TCO. Still batteries are evolving fast. Costs have dropped by 73% from 2010 until 2016 and they will keep falling in the next years [29]. According to Berckmans et al. (2017), standard lithium batteries should cost, in 2020, around 175 €/kWh [30]. We use therefore this value in our model to estimate the costs of batteries if we do not receive information from the manufacturer directly.

Lifetime of batteries are also progressing. Manufacturers used to propose warranties on batteries of five years and their length has now been extended to eight years for some of them. Lithium batteries used in normal conditions with a very high intense use (reaching 100% of depth of discharge) can hold more than 2500 cycles before they need to be replaced [31]. Assuming that electric vehicles need to be charged once a day for 260 days a year, the model considers that lithium-ion batteries should be replaced when the vehicle ages around ten years in order to be conservative. No residual value is then considered for the old battery although 80% of its energy capacity is still available. We could not find reliable values to estimate the prices of second-hand batteries.

The support for electric commercial vehicles in the Brussels-Capital Region has been replaced by another support scheme in line with the low emission zone enforcement. A subsidy is available to firms that have old vehicles and should replace them in order to access to the low emission zone. Therefore, we do not consider a subsidy in our model as that scheme does not support any more specifically electric vehicles.

The Belgian fiscal system allows a deductibility from corporate income taxes of 100% for light commercial vehicles on every cost related to the vehicle. There is no different deductibility rate between electric and conventional vehicles although it is the case in the company car regime. Hence, we assume no difference in our baseline scenario but we will test the impact of reduced deductibility rates for conventional vehicles on the competitiveness of electric vehicles. In that context, the model uses a tax rate of 24.25% on profits which is commonly used for small companies with a profit between 1 and 25,000 euros [32].

Fuel and electricity costs are assumed not to increase more than the inflation. Therefore, the TCO model does not simulate change in fuel prices given that we use the real discount rate. The prices excl. VAT is 1.22 €/l for petrol (from www.petrolfed.be for the price of "Petrol 95 RON—E10", consulted on 3rd of October, 2019), 1.27 €/l for diesel (from www.petrolfed.be for the price of "Diesel", consulted on 3rd of October, 2019), and 0.16 €/kWh for electricity (from www.brusim.be for the price for a professional customer based in 1000 Brussels, with a single rate meter and a total consumption of 10,000 kWh a year, consulted on 3rd of October, 2019).

### *2.3. Scope of the Market Research*

The supply of light commercial vehicles does not propose as many electric alternatives as the passenger car segment. Nevertheless, a total of 11 electric vehicles were selected based on the availability of the commercial information. If different versions were available, we always selected the basic model. Also, the selection paid attention to keep the diversity of the market supply by showing a range of vehicles from different vehicle categories and with different gross vehicle weight. We also include different business models with a battery renting option (Kangoo ZE-r) and a battery buying option (Kangoo ZE-b) for the same electric vehicle. To compare as accurately as possible the electric vans with their conventional counterparts, the most similar version of the selected electric vehicles was chosen. As a result, six diesel vehicles and one petrol vehicle were included in the analysis. The costs were retrieved by contacting directly the manufacturers, the distributors, the car dealers, and the regulatory bodies. Table 1 summarizes the main inputs we collected and considered in the TCO model.

**Table 1.** Overview of vehicles analyzed in the total cost of ownership (TCO) model.

| Name | Volume (m$^3$) | Gross Vehicle Weight (kg) | Purchase Price (€, VAT excl.) | Consumption (l/100 km or kWh/100 km) | Insurance (€, VAT excl.) | Range NEDC (km) | Speed Max (km/h) | Battery Capacity (kWh) | Supposed Battery Price (€, VAT excl.) |
|---|---|---|---|---|---|---|---|---|---|
| Goupil e-G4 (4 m$^3$) | 3.70 | 2100 | 29,260 | 15.0 | 588.23 | 50 | 50 | 7.2 | 6000 |
| Goupil e-G5 (4 m$^3$) | 3.85 | 2000 | 32,650 | 21.0 | 616.57 | 55 | 70 | 11.5 | 8000 |
| Alke ATX 340e (4 m$^3$) | 3.70 | 2150 | 35,200 | 14.3 | 576.17 | 70 | 44 | 10.0 | 10,800 |
| Partner D (3 m$^3$) | 3.30 | 1,980 | 15,060 | 4.70 | 791.31 | - | 152 | - | - |
| Partner P (3 m$^3$) | 3.30 | 1,940 | 15,790 | 6.20 | 927.75 | - | 174 | - | - |
| Partner E (3 m$^3$) | 3.30 | 2,175 | 30,970 | 17.60 | 759.83 | 170 | 110 | 22.5 | 4000 |
| Kangoo D (3 m$^3$) | 3.00 | 1950 | 15,750 | 4.80 | 812.30 | - | 154 | - | - |
| Kangoo ZE-r (3 m$^3$) | 3.00 | 2126 | 22,450 | 15.20 | 733.59 | 200 | 130 | 33.0 | - |
| Kangoo ZE-b (3 m$^3$) | 3.00 | 2126 | 28,950 | 15.20 | 733.59 | 270 | 130 | 33.0 | 6500 |
| D-NV300 (4 m$^3$) | 5.20 | 2780 | 24,080 | 6.80 | 870.03 | - | 158 | - | - |
| E-NV200 (4 m$^3$) | 4.20 | 2240 | 32,620 | 20.60 | 922.50 | 275 | 123 | 40.0 | 7000 |
| Crafter D (11 m$^3$) | 10.70 | 3500 | 29,761 | 8.20 | 1043.19 | - | 160 | - | - |
| Crafter E (11 m$^3$) | 10.70 | 3500 | 65,000 | 21.50 | 1027.45 | 160 | 90 | 35.8 | 6500 |
| Master D (8 m$^3$) | 7.75 | 2800 | 28,150 | 7.10 | 1027.45 | - | 148 | - | - |
| Master D (13 m$^3$) | 12.48 | 3500 | 33,150 | 7.10 | 1027.45 | - | 148 | - | - |
| Master ZE (8 m$^3$) | 8.00 | 3100 | 59,600 | 27.50 | 801.81 | 120 | 100 | 33.0 | 6500 |
| Master ZE (13 m$^3$) | 13.00 | 3100 | 63,800 | 27.50 | 801.81 | 120 | 100 | 33.0 | 6500 |
| Maxus EV80 (11 m$^3$) | 11.50 | 3500 | 57,590 | 29.20 | 985.47 | 192 | 105 | 56.4 | 16,000 |

## 3. Results

Figure 2 shows the results of the TCO for the 18 different electric vans. They are sorted from the vehicle showing the lowest TCO to the highest. Still, we isolated the three first vehicles as they belong to a specific category. Although they show a more limited speed (around 50–70 km/h) and a more limited range (about 50–70 km), these vehicles have the advantage of being very versatile. The manufacturers of these vehicles offer many different options in order to adapt them to specific customers, such as local authorities, last mile delivery companies, and industrial sites. Larger batteries could, for example, be selected in order to reach a higher range although that choice would increase the TCO. Equipment could also include a tipper, a box van or a waste collection body. It is interesting to notice that no conventional vehicles are considered in that category. Indeed, Goupil and Alke do not propose a conventional version of their electric vans although Alke used to have some models available (Alke). Such a choice might show that conventional technologies are not appropriate anymore for those types of vehicle. We isolated therefore these vehicles given that we cannot really compare their TCO with the TCO of the conventional vans we show in Figure 2. These vehicles are also not considered in the sensitivity analysis as the goal of that section is to identify break-even points of electric vehicles with their similar versions. Still, we integrate them in our analysis as these vehicles are interesting to be considered as an alternative to conventional vans.

The next 15 vehicles show all similar performances in terms of speed and the different electric vans show all a range above 100 km. They vary in terms of payloads and volumes. The gross vehicle weight starts at around 2000 kg and ends at 3500 kg. Volumes start at 3 m$^3$ and end at 13 m$^3$. The van showing the lowest TCO is the petrol version of the Peugeot Partner. Compared to its diesel version, the second vehicle with the lowest TCO, Figure 2 shows that the petrol van benefits especially of lower maintenance costs and lower purchase costs than its diesel version. Indeed, depreciation rates of petrol vehicles are falling less fast than diesel vehicles, probably given the coming regulations that target especially diesel vehicles. As a result, these advantages of the petrol vehicles offset their higher fuel consumption and their higher insurance costs compared to diesel vehicles. That analysis is striking given the dominance of diesel in the light commercial vehicle segment. Preferences for petrol vans might therefore become higher than for diesel vans, as we can observe in the car segment in Belgium.

Still, the electric vehicle can offer another alternative to diesel vehicles in the segment of small vans. Indeed, Figure 2 shows that the Partner electric, the Kangoo ZE-r (with a battery renting system) and the Kangoo ZE-b (with a battering buying system) have all a TCO that is quite comparable to their diesel counterpart (the diesel Kangoo and the diesel Partner). We observe a TCO difference of less than 1% between these similar models although their cost structures are quite different. Purchase costs of electric vehicles are typically higher than their conventional versions despite a lower depreciation rate for electric vehicles. Electric vehicles are also entailed by the extra costs of their batteries. Still, these costs are balanced by the lower insurance, maintenance, and fuel costs of electric vehicles compared to their diesel versions, which results in a similar competitive position between electric and diesel vehicles in that segment. The sensitivity analysis will however be able to better show under which conditions electric, diesel, and petrol technologies perform based on their TCO.

Looking at the medium sized vans, Figure 2 shows that the electric Nissan NV200 benefits from a much lower TCO than the diesel NV300. We observe a difference of almost 15%. The extra battery costs of the electric NV200 are covered by its lower maintenance and fuel cost compared to the diesel version. In addition, the NV200 benefits from a lower purchase price (excluding the battery costs) compared to the NV300. This lower purchase price might be explained by the slight model difference we considered. Indeed, the diesel NV200 is not manufactured anymore and has been replaced by the next generation of that model, the NV300. The comparison is therefore less straightforward although we can assume they are close enough to be compared. On the one hand, the new version might be more costly as it is a new model generation. On the other hand, the consumption of the old diesel NV200 might have been higher which could have balanced the lower purchase costs of the old diesel NV200.

The last segment includes the Master of Renault, The Crafter of Volkswagen, and the EV80 of Maxus. They all show a gross vehicle weight of more than 3000 kg and have volumes above 8 m$^3$. These are the largest vans before we switch to the truck segment. Figure 2 shows that the TCO of diesel vehicles are in this segment clearly different from the TCO of electric vehicles. Comparing the different versions together, we can see that the TCO of electric vans are higher by around 15% than their diesel versions. This difference is mainly explained by the much higher purchase cost of electric versions (knowing that we have extracted the estimated battery costs from the purchase price). The more limited economies of scale achieved on the expected lower sales of the electric versions compared to diesel versions could perhaps explain that cost difference between electric and conventional versions. Still, Figure 2 shows that the EV80 proposes a comparable purchase cost to similar diesel vehicles. Its TCO remains however higher than its diesel counterpart given the larger battery costs of that model. These extra battery costs compared to the other electric vans of that segment are due to the larger battery capacity proposed in the EV80. The battery has a size of 56.4 kWh which offers a higher range than the battery of 33 kWh proposed in the electric Master and 35.8 kWh proposed in the electric Crafter. Since Maxus does not propose a smaller battery, the TCO of the EV80 cannot be reduced. As a result, the competitive position of the different electric vehicles remains more difficult in the segment of large vans.

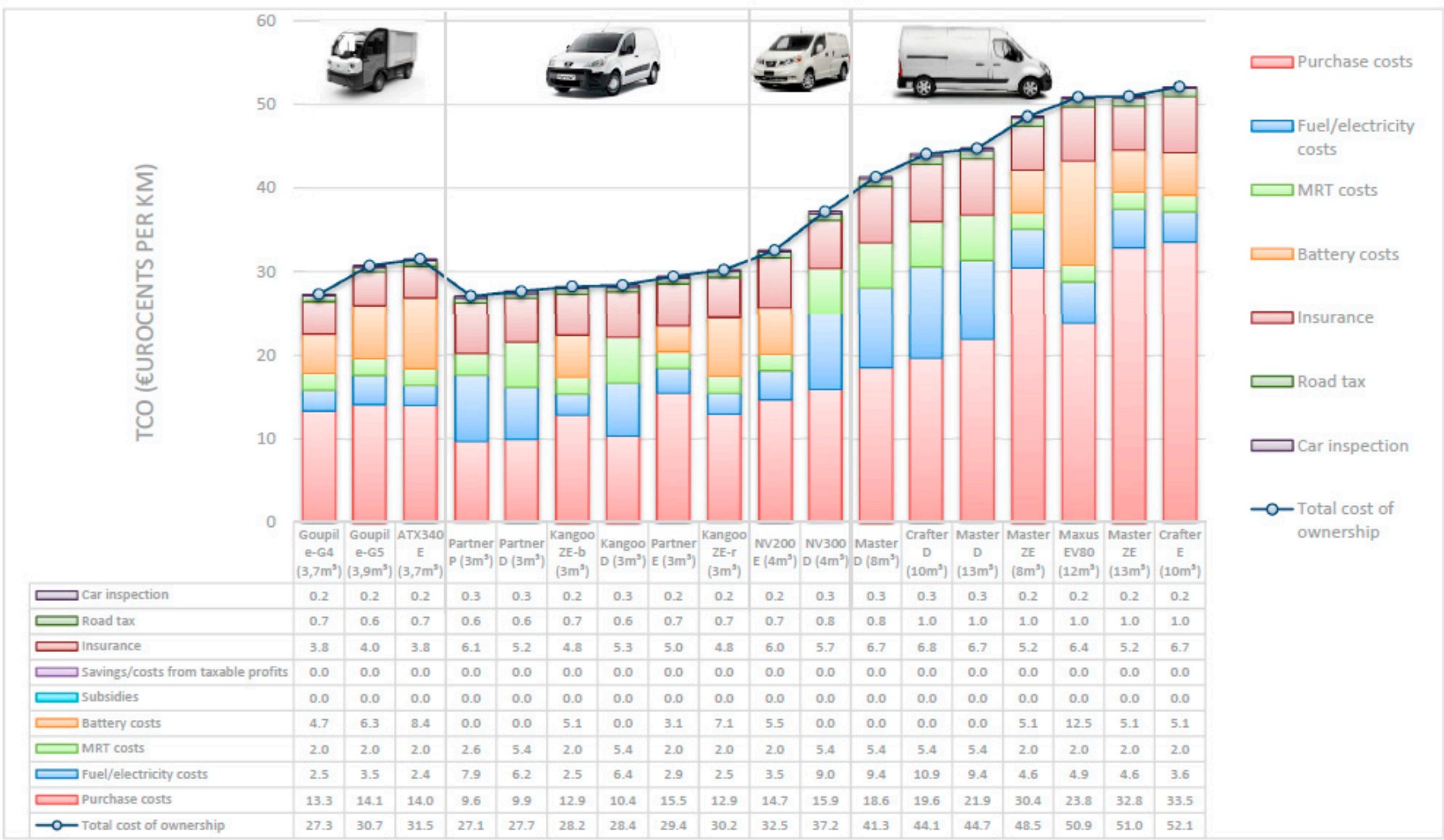

**Figure 2.** Results of the total cost of ownership analysis.

## 4. Discussion

Different assumptions of the TCO model are investigated in this sensitivity analysis. We explore in the following sections the impact on the results of the TCO of the kilometers driven, the ownership period, the residual value of the battery, and two different types of policies (fiscal incentives and exemption from a kilometer-based charge). Based on this sensitivity analysis, we identify the break-even points where electric vans become more competitive than their conventional counterparts. Hence, we did not include the electric G4, G5, and Alke in our sensitivity analysis as they do not have conventional versions.

### 4.1. Kilometers Driven

The results shown in Figure 2 have assumed an average distance of 16,000 km per year. Figure 3 shows the sensitivity of that criteria on the TCO of the light commercial vehicles we have analyzed. The trend is clear and straightforward: the more the vehicle drives, the more the TCO per kilometer decreases. However, the TCO of electric vehicle drops faster than the TCO of conventional vehicles given its lower running costs. Conversely, we can see that, when distances are low, the competitive gap between electric vehicles and their diesel version is larger.

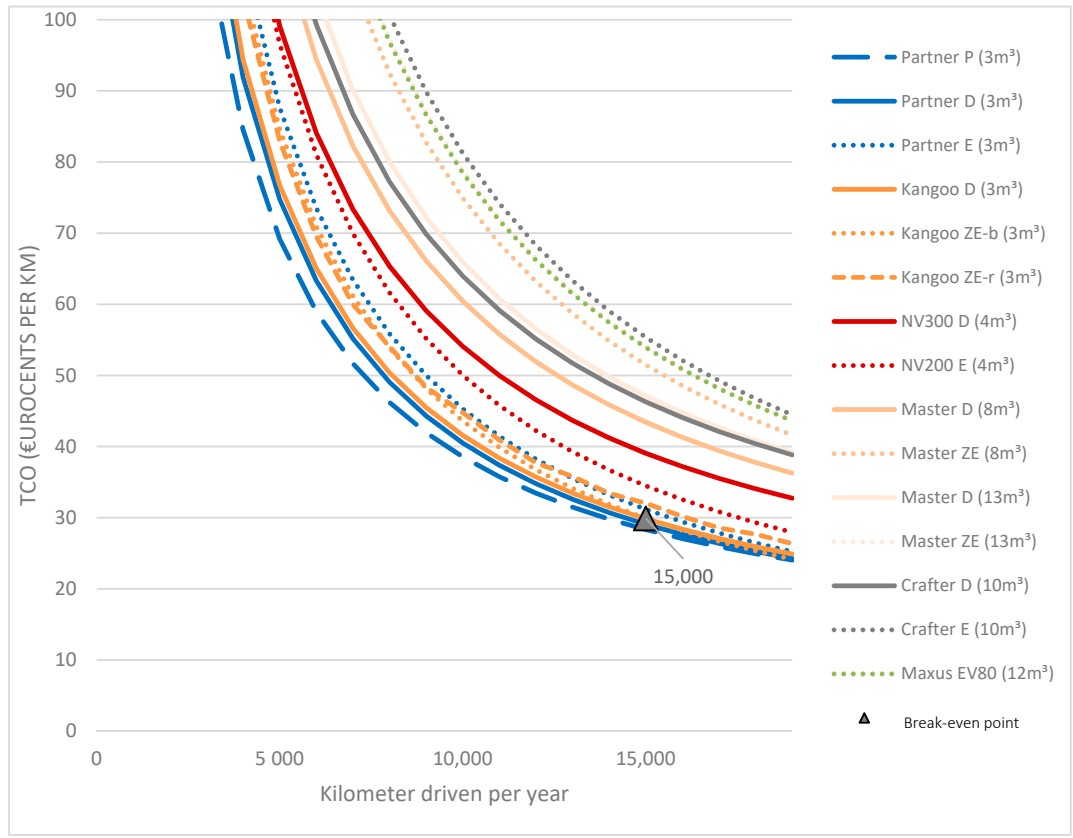

**Figure 3.** Sensitivity of the kilometer driven on the total cost of ownership (TCO) results.

Still, the competitive position of electric vans remains quite stable in this analysis as we do not observe many break-even points. We can identify in Figure 3 only one break-even point between the electric Kangoo ZE-b and the diesel Kangoo. The electric version starts to have a lower TCO once the distance driven is higher than 15,000 km per year. We found two other break-even points but they are located at the extremes. The electric NV200 starts to be more competitive than its diesel counterpart already when a minimum distance of 3000 km per year is driven. The petrol Partner becomes less interesting than its diesel version when more than 20,000 km are driven per year.

### 4.2. Years of Ownership

The results of the TCO assumed an ownership of 8 years. However, Figure 4 shows that the TCO of electric vans is sensitive to that parameter, particularly during the first years of ownership. Indeed, we assumed that batteries have no residual values as we could not find reliable values to estimate the prices of second-hand batteries. It is risky to purchase a second-hand battery given that we do not know how the battery was managed. Still, we test the sensitivity of that parameter as this assumption can be considered quite conservative. As a result of that assumption, selling an electric vehicle after one year would generate in an important loss given the share of battery costs in their TCO. With a longer period of ownership though, the TCO of electric vans drops faster than the ones of diesel vehicles as they benefit of their lower running costs. They can compensate during a longer time their high purchase costs with their low operating costs. In that extent, the sensitivity analysis on the period of ownership is similar to the sensitivity analysis on the kilometers driven given that more years of ownership involves more kilometers driven.

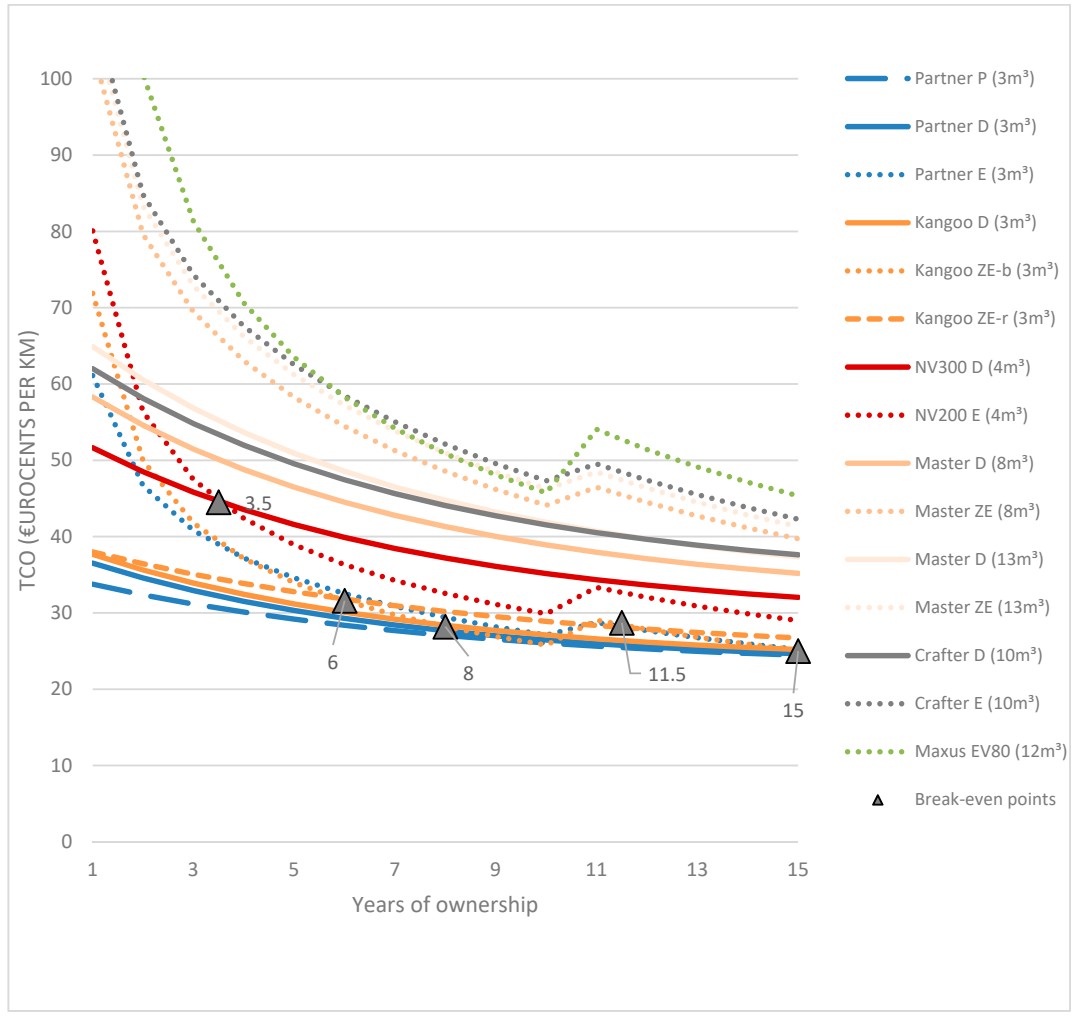

**Figure 4.** Sensitivity of the years of ownership on the TCO results.

Still, Figure 4 shows more break-even points than in Figure 3. We find again that the NV200 becomes quickly more competitive than its diesel counterpart with a break-even point at 4 years. The Kangoo ZE-b is an interesting case as it experiences several break-even points. Its TCO becomes first lower than the Kangoo ZE-r after 6 years of ownership. From that point, buying the battery offers a more interesting TCO than keep renting the battery. However, if the battery must be replaced, the TCO of the Kangoo ZE-b increases while the TCO of the Kangoo ZE-r keep decreasing. The costs of such a

replacement are covered by the manufacturer in the renting option of the battery (Kangoo ZE-r), but not in the buying option of the battery (Kangoo ZE-b). As we assume the battery is to be changed after 10 years, Figure 4 shows that in year 11, the Kangoo ZE-b shows a higher TCO than the Kangoo ZE-r. Already in year 12, the option of buying the battery becomes more interesting again. We find two similar break-even points between the Kangoo ZE-b and its diesel version. The TCO of the electric vehicle becomes lower than its diesel version a first time after 8 years of ownership and a second time after 15 years of ownership due to the battery replacement of the Kangoo ZE-b.

Figure 4 highlights well the effect of the battery on the TCO of electric vehicles. As we assume that no residual value can be captured from the battery, the TCO of electric vans is particularly affected in the first years after a battery purchase. The TCO of electric vehicles can therefore be optimized by using the vehicle until the battery has to be replaced. This optimization does not allow however the heavier electric vans to reach the TCO of their diesel versions.

### 4.3. Residual Value of the Battery

Given the important impact of batteries on the TCO, the assumptions regarding the residual value of the batteries can be critical on the results. Indeed, we considered that batteries do not have a residual value. Batteries has however a remaining capacity, with a minimum of 80% when they need to be replaced for automotive applications. Residual value could therefore potentially be captured by second hand applications of batteries.

Figure 5 shows the sensitivity of that criteria on the TCO results. The larger the battery, the more that criteria impacts the TCO. Indeed, the Maxus EV80 have the largest battery among the vehicles that we considered. A residual value of 50% on the battery after 8 years could already reduce the TCO of that vehicle by 14% which would result in a similar TCO with its diesel counterparts. The other electric vehicles of the large vans segment cannot however reach a break-even with their diesel versions, even with a theoretical residual value of 100%. Their batteries are smaller and the effect of that parameter on the TCO is therefore more limited. Still, the competitive gap is reduced from a TCO difference of about 15% to a difference of maximum 5%.

Given that the competitive gap is smaller within the smaller vans, some break-even points are still identified despite the more limited impact of that parameter on small batteries. Since the electric Kangoo ZE and NV200 have already a lower TCO than their diesel versions when there is no residual value on the battery, the last electric van to be investigated in the smaller segment is the electric Partner. The break-even with its diesel version occurs when a residual value of 50% can be retrieved after 8 years. If the residual value rises to 70% after 8 years, then the TCO of the electric Partner becomes also lower than its petrol version.

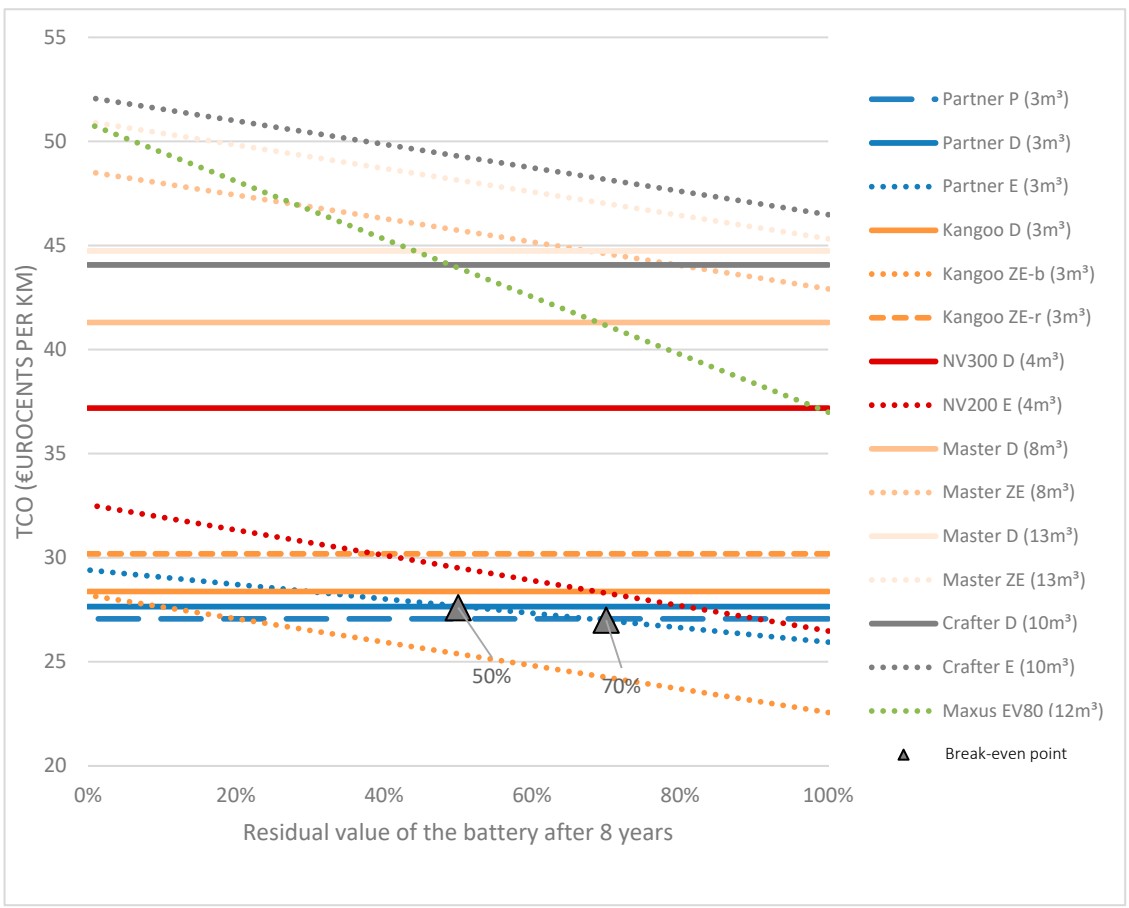

**Figure 5.** Sensitivity of the battery residual value on the TCO results.

### 4.4. Fiscal System

It is also interesting to conduct a sensitivity analysis on criteria that are policy dependent. We can, on the one hand evaluate the impact of future policies on the competitiveness of electric vans in Brussels. On the other hand, we can extrapolate the results of the TCO to cities that have a different policy context.

In Figure 6, we evaluated the sensitivity of our results to different fiscal systems where the deductibility of costs related to conventional vans would be reduced while the ones of EV would remain at 100%. The analysis shows that the impact of such a measure can be effective on the competitiveness of electric vans. We observe indeed several break-even points. The first electric vehicle reaching a break-even point starts with a deductibility of maximum 76% on costs related to conventional vehicles. Then the electric Partner and the electric Kangoo ZE-r become more competitive than their diesel versions. The electric Partner becomes more competitive than its conventional versions later, when the deductibility is further reduced to 68%.

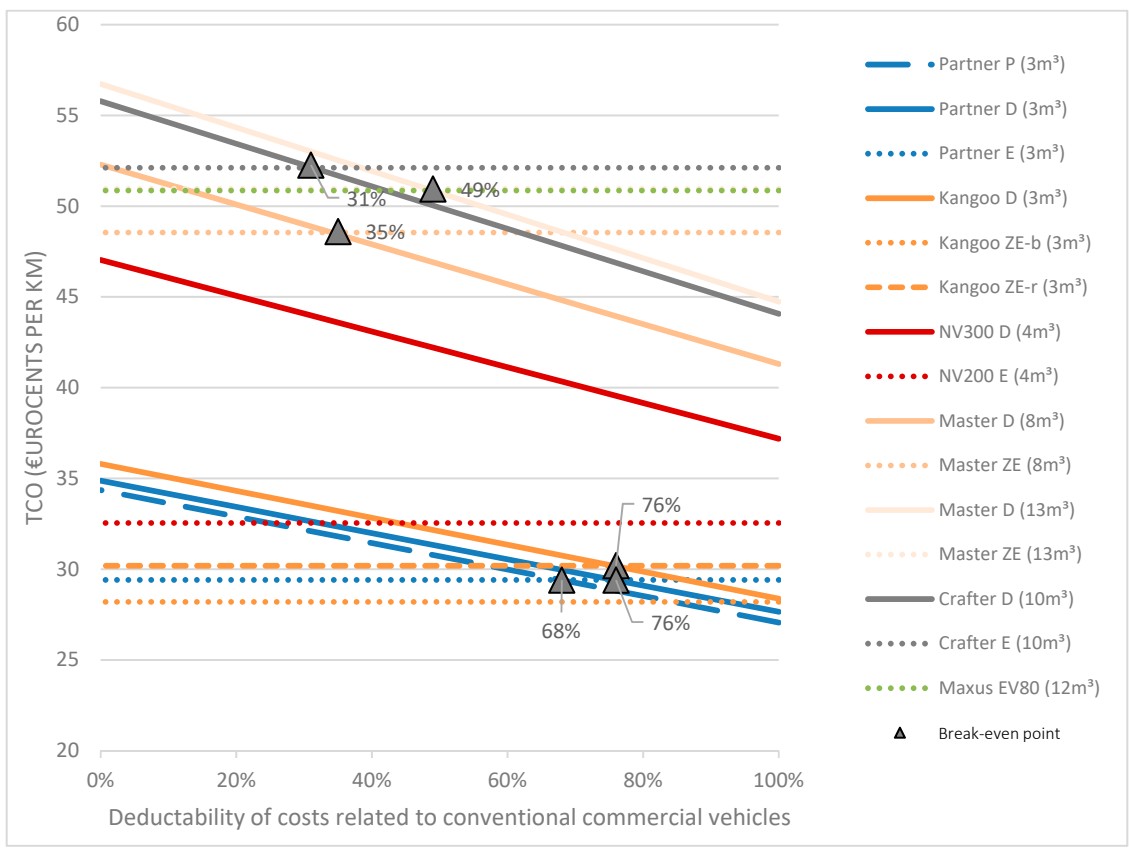

**Figure 6.** Sensitivity of the fiscal system on the TCO results.

The effectiveness of that measure is however especially remarkable in the heaviest segment of vans. Indeed, Figure 6 shows that the electric Master ZE of 13 m$^3$ and the EV80 reach both a break-even point with the diesel Master 13 m$^3$ when deductibility of costs related to conventional vans are reduced to 49%. If the deductibility is further reduced to 30%, then every electric van considered in our analysis become more competitive than their conventional counterpart. As a result, we can identify the fiscal tool to be a powerful instrument. By reducing the fiscal incentives towards conventional vans, policy makers can stimulate the competitiveness of electric vans.

*4.5. Kilometer-Based Charge*

At a more local level, it is possible also to introduce an urban toll, with an exempt on electric vehicles. Figure 7 shows the sensitivity of a kilometer-based fee on the results of the TCO. Since electric vehicle are not affected by this analysis, their TCO remains constant. The TCO of conventional vehicles increase however with higher taxes.

The results show here several break-even points. The TCO of the electric Partner becomes lower than its petrol and diesel versions when the kilometer charge is higher than, respectively, 1.75 eurocents and 2 eurocents per kilometers. With a tax of 2 eurocents per kilometer, electric vehicles could therefore already be more competitive in the small van segment. The kilometer-based charge should increase to 6 eurocents per kilometer in order to see the electric Master ZE 13 m$^3$ more competitive than its diesel version. The electric Master ZE 8 m$^3$ has more difficulties to compete as its TCO becomes lower than its diesel version later, when the charge rises to 7 eurocents per kilometer. Finally, the electric Crafter reaches its break-even point with its diesel version when the kilometer charge rises to 7.5 eurocents per kilometer.

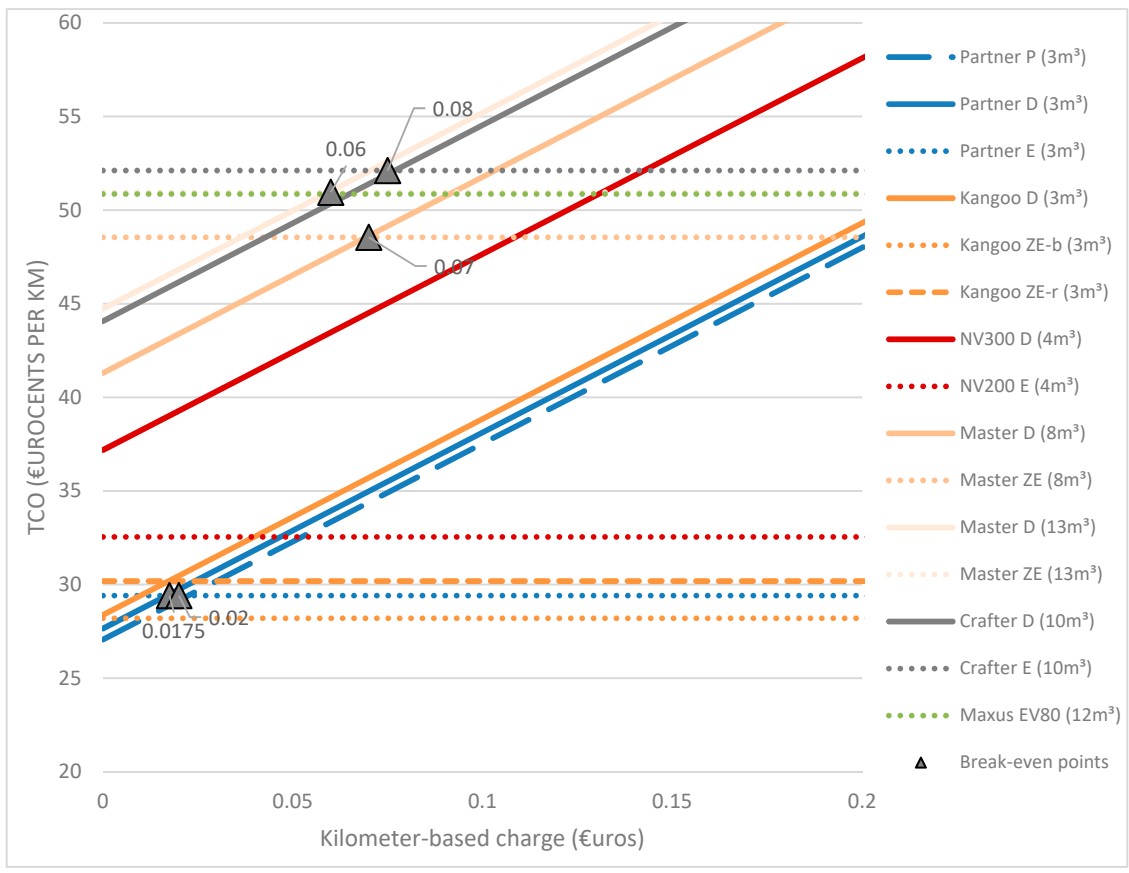

**Figure 7.** Sensitivity of a kilometer-based charge on the TCO results.

## 5. Conclusions

The results of this total cost of ownership analysis have shown that the competitive position of electric vehicles is still difficult today in the segment of light commercial vehicles. Small electric vans do compete with their conventional versions as they weigh less and need therefore smaller and less expensive batteries. The heavier the electric vehicle becomes however, the more difficult their competitive position become with their conventional versions. There are therefore few financial incentives to switch from conventional to electric vans.

The paper has explored the sensitivity of a few critical assumptions. The results have shown that the competitiveness of electric vans can be improved by optimizing the usage of the vehicle. By using intensively the vehicle, the TCO of electric vehicles drops quicker than the TCO of conventional vehicles as it benefits from low running costs. Still, it is difficult to use intensively an electric van given the limited range of those vehicles. Ownership time seems therefore a more relevant criteria to optimize. First, it has a more important impact on the TCO, especially in the first years. Second, the sensitivity analysis showed the critical impact of battery replacements on the TCO of electric vans. As a result, we find that the ownership of the vehicle should last until the battery has to be replaced to minimize the TCO of electric vans. That conclusion is drawn from the assumption according which the batteries have no residual value. However, if value can be captured from used batteries, the TCO of electric vans can be further reduced, especially for heavier vans.

The contribution of the paper lies especially in the policy recommendations to support the competitiveness of electric light commercial vehicles. The potential of a more sustainable fiscal system and of a kilometer-based charge were assessed. Both were found to be very effective. If fiscal incentives for conventional vans are reduced, we found that all the electric vans can become more competitive than their conventional versions, even in the heaviest segment of vans. Policy makers have therefore a

powerful measure to support the electrification of city logistics. The deductibility of costs related to conventional vans should be reduced to at least 30% in order to do so. Another powerful measure that was found is the introduction of a kilometer-based charge. In order to make all electric vans more competitive than their conventional versions, a charge of at least 7.5 eurocents per kilometer should be introduced on conventional vans.

The paper highlights therefore the gap between the ambitious climate objectives and the limited supporting policies. Indeed, the support of electric vans in the Brussels-Capital Region has been limited. Subsidies have even been canceled. The paper showed, however, that policy makers have powerful tools to support the electrification of city logistics. As the supply of electric vans keeps growing (the Mercerdes Vito and the Iveco Ecodaily are coming soon), measures should be implemented in order develop the potential of that technology and close the gap with the climate objectives.

**Author Contributions:** Writing—original draft, P.L.; Writing—review & editing, C.M. and J.V.M.

**Funding:** This research received no external funding.

**Conflicts of Interest:** The authors declare no conflict of interest.

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
