# Peer review of "How to Improve the Total Cost of Ownership of Electric Vehicles: An Analysis of the Light Commercial Vehicle Segment"

_wevj, doi:10.3390/wevj10040090_

Round 1

Reviewer 1 Report

The papers analyzes how the TCO of electric light commercial vehicles changes on the basis of a number of parameters that include, amongst others, the annual distance travelled, the period of ownership, the residual value of the number of kilometers driven, fiscal incentives and the period of ownership. Results show that both a kilometer-based charge and reduced fiscal incentives for conventional vans may lead to an increase in competitiveness for electric commercial light duty vehicles.

I believe that the paper is well-written and deals with an interesting topic in a clear and synthetic manner. Yet, I also believe that the publication could also be enriched by considering that the TCO of the electric vans may change according to a number of aspects that has been not included in the study, such as the the percentage of urban/suburban trips, and the effect of different energy consumption depending on the weather.

Overall, I believe this contribution deserves publication after a set of minor revisions the Authors I invite to undertake.

Minor comments

Page 1, line 39. I believe that there is no need to place a minus in front of 60%.

Page 3, line 13. I believe you should provide a source that backs the data reported in this part of the sentence “…average period of ownership of light commercial vehicles was 8.2 years in 2016.”

Page 3, line 33. What does “FEBIAC” stand for? Please define it, since it is the first time you use it in the text. You should also place the year to which this publication refers to (2009)

Page 4, lines 6 to 9. It seems awkward that you are not able to find information about maintenance for the Belgium case and you have to resort the UK example. I am wondering how this could affect your results. Have you tried to interview a sample of mechanics who had experience in this respect?

Page 5, lines 1-3. This is fine, but you may consider to add a scenario considering possible evolutions in the fuel/electricity price according to change in the price of oil and taxation.

Page 6, Table 1. Maybe it would be useful to place a name for the subgroups of vans described. This would help the reader to better connect the text with Figure 2.

Page 7, line 31. I believe that there might be a mistake here. It should be "quite" and not "quiet". It is a typo, that appears also in two other instances.

Page 8, line 57. Maybe there is a preposition missing, such as "which" or "that".

Page 8, line 86. Here should be "quite" and not "quiet" as similarly reported above.

Page 8, line 90-91. You state that “The electric NV200 starts to be more competitive than its diesel counterpart already when a minimum distance of 1,000km per year is driven.” Are you sure that is it 1,000km per year and not 10,000km per year?

Page 9, line 102. Here should be "quite" and not "quiet" as similarly reported above.

Page 9, lines 105-106. You state “hey can compensate during a longer time the high purchase costs of electric vehicles with their low operating costs”. Maybe there is no need to repeat “electric vehicles”.

Page 13, line 197. I believe that you should write “vans” instead of “van”.

Page 14, lines 219-220. It would be interesting and helpful if you could be more precise in suggesting exactly which policies should be enacted. Moreover, I would expect would also expect some recommendations for car manufacturers.

Author Response

Thank you for your very useful feedback and constructive comments.

Your comment on the impact of energy consumption is interesting as we wanted to integrate it in our sensitivity analysis. However, we decided to leave it out as it did not have a key influence on the TCO. It has of course an influence, but it is not in the top 5 criteria. As we want to focus the paper on the most sensitive criteria, we did not include an analysis on the percentage of urban/suburban trips and on the effect of different energy consumption depending on the weather.

We integrated well your other suggestions. We reply below to some of your comments.

Page 4, lines 6 to 9. It seems awkward that you are not able to find information about maintenance for the Belgium case and you have to resort the UK example. I am wondering how this could affect your results. Have you tried to interview a sample of mechanics who had experience in this respect?

>> The figures we used from the UK make sense in Belgium as well. We checked it indeed. But that paper from the UK was the most reliable source we could find to estimate the maintenance costs of the three different technologies from the same source. Anyway, as this reference shows, maintenance costs remain stable across countries. So if we choose another country considered in that reference, it would not have changed much. But selecting UK figures seemed more relevant than selecting Japanese figures for an analysis in Belgium.

Page 5, lines 1-3. This is fine, but you may consider to add a scenario considering possible evolutions in the fuel/electricity price according to change in the price of oil and taxation.

>> Indeed, it is an interesting scenario to consider. We had tested it but we did not keep it as we did not find that criteria to be a key sensitive parameter.

Page 6, Table 1. Maybe it would be useful to place a name for the subgroups of vans described. This would help the reader to better connect the text with Figure 2.

>>The table is already dense. We did not have space to add an extra column. Instead, we have added intermediary lines to better identify the segments.

Page 14, lines 219-220. It would be interesting and helpful if you could be more precise in suggesting exactly which policies should be enacted. Moreover, I would expect would also expect some recommendations for car manufacturers.

>> The paper is oriented towards policymakers. The policies we recommend are detailed in the paragraph just above lines 219-220. We show the level policymakers should reach in order to make the TCO of e-vans more attractive financially.

Reviewer 2 Report

This is a very interesting work and aims to analyze the total cost of ownership of electric light commercial vehicles. Even though the proposed method does not present a high level of novelty, the gained results provide important insights of using electric light commercial vehicles. The authors explore the sensitivity of a few critical assumptions in order to present an up-to-date overview of the competitive position of electric vehicles in the light commercial vehicle market of the Brussels region. A further analysis of South Europe countries would be interesting to compare results in different territorial domains. Overall, the paper is well-written and very easy to understand.

Author Response

Thank you for your very positive feedback. We take with us your suggestion for a future research. We have been through the paper to catch the last English mistakes.

Reviewer 3 Report

In their paper “How to improve the total cost of ownership of electric vehicles? An analysis of the light commercial vehicle segment” the authors set out to present an update of a TCO comparison between electric vans and their and combustion engine equivalents. The authors present updated input parameters, for example purchase prices, the discount rate and battery lifetime expectations to carry out the TCO. They compare and discuss the results with a sensitivity analysis that includes aspects such as policy measures. The author’s main finding is that certain policy measures can have a high impact on the TCO and therefore are an important instrument to support the adoption of light commercial electric vehicles.

The topic of the paper still is relevant and as the authors point out, a complete updated TCO of light EVs that reflects the recent updates is interesting. As a general remark, the authors describe and discuss a solid and detailed model for TCO calculation. However, currently the paper has deficiencies especially with regard to discussing the state of the art and consequently the authors fall short of clearly positioning their findings in relation to the current literature and underpinning their contribution and to the body of knowledge. 

This issue becomes especially relevant when checking the references. Out of 23 references, only 10 references are peer reviewed journal or conference papers. Out of these ten papers, seven are authored or co-authored by the authors of this paper. The remaining three references include one paper about the method; a second one is from 2009. Such extensive self-citations while not taking into account many of the past and current relevant papers does not present a good scientific practice. Regarding this point, a meticulous rework is required.

Furthermore, language, phrasing and many small details require the attention of the authors, in order to publish the paper. Grammar, tables, figures and little issues in the references leave the impression of a hastily written paper. Thus, I recommend a rework followed by a professional proofread before resubmission. I will point out several issues of this overall interesting paper below and hope that my recommendations help to improve the quality of this paper.

Abstract

The abstract is concise and stands for itself. Small issue:

First sentence: “This paper analysis…” should this be “This paper analyses…”?

Introduction
The motivation described clearly, to address the above mentioned concerns please consider to add some paragraphs about the state of the art, or introduce a chapter “State of the Art”.

Two of the given three references on TCOs for EVs refer to your own work, the third of the references to a TCO on EVs in the passenger car segment.

Here are some examples of TCO calculations, most on commercial electric vehicles, many of them also discussing the impact of policy. These and more papers could be considered:

Camilleri & Dablanc (2018) An assessment of present and future competitiveness of electric commercial vans. Levay, Drossinos & Thiel (2017) The effect of fiscal incentives on market penetration of electric vehicles: A pairwise comparison of total cost of ownership. Taefi, Stuetz & Fink (2017) Assessing the cost-optimal mileage of medium-duty electric vehicles with a numeric simulation approach. Feng & Figliozzi (2013) An economic and technological analysis of the key factors affecting the competitiveness of electric commercial vehicles: a case study from the USA market Lee, Thomas & Brown (2013) Electric urban delivery trucks.

Figure 1 is useful to show the trend over time. Small correction: the Y-Axis should be labelled and the numbers fractions are written with a comma: 0,04% . It should be a decimal point instead: 0.04%

Methods
You describe the method elaborately. Small issues are:

2.2 Discount rate: You use a direct quote “the rate of interest…”, hence please add the page number in the reference, for example [8, p. 5]. Page 3 lines 13-15: the content is duplicated in lines 34-37. Consider removing the duplication. Page 4 line 6: mixing the quotation styles seems unusual “[18]. Palmer et al (2018)”. Consider harmonizing. Page 4 line 8: “We used … “ the sentence seems somewhat mixed-up. Page 4 line 9: “… we assume a maintenance costs…” is mixing singular and plural. Page 4 line 9: “306€”: there should be a space between the value and the unit. Please check the rest of the paper as well (page 6 13m^3 requires a space as well, etc.). Page 4 lines 23/24: … “Manufacturers used to propose warranties on batteries of 5 years and it has now been extended). The sentence sounds weird (“it?”. Same sentence: single-digit numbers should be spelled out (five instead of 5, eight instead of 8). Also check the remainder of the paper for these style issues (i.e. page 5, lines 13) Page 4 line 28. When a vehicle is charged once per day for 260 days a year, this adds up to 2600 cycles. You quote that [22] found that a battery can hold more than 2500 cycles. This is less than the assumed 2600 cycles. Why do you suggest your assumption of 2600 cycles is conservative (line 28)? Page 5 line 5: “… as many electric alternatives then …” should be “…as many electric alternatives as…” Page 5 lines 11/12 “To be able to….” and line 14 “The costs considered…” have a very ‘french’ style of placing words. There are many more examples in the paper that I will not list here. A proofreader would capture these. Table 1 is useful to give an overview on the compared models, but has several issues: First line: Replace the only abbreviation “GVW” by the full term, there should be enough space. First line: ”Speed Max”: max is capitalized, but “Battery capacity” capacity is not. “Supported Battery price” is again mixing capital and small letters at the beginning of the word. Please harmonize. The text in the columns is centered. Right aligning all columns would make them easier readable, especially if in combination with the next point The columns “Volume”, “Consumption” and “Battery capacity” contain values that have none, one or two digits behind the comma. Harmonizing the number of digits after the comma would create a view that is easier to read. The values in the table sometimes use a decimal point and sometimes a comma to indicate the decimal place. Use the decimal point for all of them. Page 5, all footnotes: “consulted” is capitalized after the comma, while it should be a small letter. By the way, footnote 5 misses a space between the value and unit (10,000 kWh)

Results
The results are well presented. However, sometimes there are speculations mixed in-between the results, for example:

Page 6 line 11: “Such a choice might show that…” Page 5 line 24: … probably given the coming regulations ….” Page 8 line 26: “… could perhaps explain the cost difference…”

Please keep the results section very focused on the pure and objective direct results that any researcher would derive by applying the method. Please move any explanation of the results to the discussion section.

Small issues:

Page 8 line 54 “The” should not be capitalized Page 8 line 56 “… before we switch .. “ sound somehow very colloquial

Discussion:
The findings are well discussed, especially the sensitivity analysis offers interesting insights. The only major shortcoming has been addressed before: The section does not contain to a discussion of the results in relation to the state of the art. It would be very interesting if other authors have come to similar conclusions as you (some have), or in case not why. Especially:

In relation to the findings about policy options. In relation to chapter 4.4: Reducing fiscal incentives of conventional vans might may even reduce the TCO of larger vans to a point where they become competitive – is this what do the studies about the next vehicle size, the medium duty commercial vehicles also find? A kilometer based charge is an efficient policy measure, especially for larger vans (again compare to the literature on medium duty vans)

Some minor issues:

Page 8 line 79: “Kilometre driven” vs. “kilometer” everywhere else in the text. Page 8 line 87: “break-even” vs “Break Even Point” without the hyphen and capitalized in Figure 3. Page 10 line 132: “capacity remaining in the battery”. Capacity does not remain in a battery; a battery has a remaining capacity. Page 11 line 138: “The more the vehicle has a large battery”. Use: “The larger the battery, the…” Page 11 line 145: “competitive gap is thinner” sound unusual. Consider “smaller”, or another word. Page 13 line 188: “7,5 “ remove comma, add decimal point “7.5”

Conclusion
The conclusion briefly summarizes the findings implications of the findings well. The implications for policy makers are pointed out clearly.

However, the discussion of the contribution to knowledge could be somewhat more elaborate. The study set out to update the competitive position of commercial EVs in the Brussels-Capital. Thus, in order to answer the overarching research question, it would be interesting to briefly, but explicitly, point out the differences between your study from 2015 and today. For example, one of your previous studies already found that the ownership of the vehicle should last until the battery has been replaced. Just to be on the safe side of self-plagiarism, point out that the result mentioned on page 14 in line 202/203 is no new finding, but the current study confirms earlier findings.

Finally, pointing out the limitations of the study would add to the scientific soundness.

References:

Please carefully check all references, there are some minor issues.

Some lines end with a full stop, some not. Reference 4 uses an abbreviation (VUB). Please spell out and complete the reference, i.e. with a publication date. Reference 13: Is there a publisher? Reference 16: Add a space in from of “Rapport”

Author Response

Thank you for your very useful feedback and constructive comments.

We took the time to integrate the state of the art and update our references. Thanks for sharing the different papers. These were however mainly focused on the truck segment. We show now better that the state of the art of the TCO of light commercial vehicles is limited. As a result, the paper is better positioned on a research gap.

A few additional replies to your comments.

You quote that [22] found that a battery can hold more than 2500 cycles. This is less than the assumed 2600 cycles. Why do you suggest your assumption of 2600 cycles is conservative (line 28)?

>> 260 days times 8 years of ownership makes less than 2500 cycles. This way, we feel conservative.

Page 4 line 6: mixing the quotation styles seems unusual “[18]. Palmer et al (2018)”. Consider harmonizing.

>>Agree that this is not well harmonised. Still we kept this way as we want to stress the reference here. It is easier to use it an harmonised way with other styles but we need here to respect the guidelines of the journal.

The results are well presented. However, sometimes there are speculations mixed in-between the results, for example:

Page 6 line 11: “Such a choice might show that…” Page 5 line 24: … probably given the coming regulations ….” Page 8 line 26: “… could perhaps explain the cost difference…”

Please keep the results section very focused on the pure and objective direct results that any researcher would derive by applying the method. Please move any explanation of the results to the discussion section.

>>We understand your point. We usually do that in the discussion section indeed. But we want to keep the discussion structured around 5 key points. And these are not included in one of these points. Plus these “speculations” contribute to understanding the results. Sometimes we feel it is relevant to give a little explanation to make the text easier to understand than giving only hard figures to the readers.

Round 2

Reviewer 3 Report

The authors addressed all major concerns, or explained their position. The current version of the document is hence nearly fit for publishing.

However, in the best interest of the journal and the authors, I'd highly recommend a professional proof-read before publishing the report. The currently uploaded version contains formulations that require improvement. Examples:

> Page 20, line 16 / 17: "Costs of the closest country was therefore considered, the UK."

> Page 31, lines 108/109: "Batteries has however a remaining capacity, with a minimum of 80 % when they need to be replaced for automotive applications." 

These are just two arbitraty examples, not the only two issue. Please let the full text be checked.